# A Unified Debiasing Approach for Vision-Language Models across Modalities and Tasks

**Hoin Jung, Taeuk Jang, Xiaoqian Wang**[*]
Elmore Family School of Electrical and Computer Engineering
Purdue University
West Lafayette, IN 47907
{jung414, jang141, joywang}@purdue.edu

## Abstract

Recent advancements in Vision-Language Models (VLMs) have enabled complex multimodal tasks by processing text and image data simultaneously, significantly enhancing the field of artificial intelligence. However, these models often exhibit biases that can skew outputs towards societal stereotypes, thus necessitating debiasing strategies. Existing debiasing methods focus narrowly on specific modalities or tasks, and require extensive retraining. To address these limitations, this paper introduces Selective Feature Imputation for Debiasing (SFID), a novel methodology that integrates feature pruning and low confidence imputation (LCI) to effectively reduce biases in VLMs. SFID is versatile, maintaining the semantic integrity of outputs and costly effective by eliminating the need for retraining. Our experimental results demonstrate SFID's effectiveness across various VLMs tasks including zero-shot classification, text-to-image retrieval, image captioning, and text-to-image generation, by significantly reducing gender biases without compromising performance. This approach not only enhances the fairness of VLMs applications but also preserves their efficiency and utility across diverse scenarios. The code is available on GitHub.

## 1 Introduction

Vision-Language Models (VLMs) have revolutionized the way we handle multimodal tasks by enabling simultaneous processing of text and image data. Models such as CLIP [31] and XVLM [43] serve as foundation models [41] for various downstream tasks demonstrating the remarkable versatility of these systems such as zero-shot classification and text-to-image retrieval. Additionally, models like BLIP [25] and CoDi [37] enhance this spectrum by facilitating tasks such as image captioning and text-to-image generation. These examples highlight the diverse capabilities of VLMs in adapting to various specific multimodal interactions.

Despite their remarkable capabilities, VLMs have a critical bias issues, often skewing the model outputs in ways that reflect societal stereotypes [21] such as gender [46] or racial [45] biases in assigning professions or describing scenarios. For example, studies have identified biases in multi-class zero-shot classification [17], where models might disproportionately associate certain professions with specific genders. Similarly, biases in text-to-image retrieval [18, 39] can lead to the preferential retrieval of images that reinforce stereotypical narratives. The implications of these biases extend to image captioning [45, 20] and text-to-image generation [10], where the descriptive and generative capacities of VLMs may perpetuate and even amplify existing societal prejudices. These issues highlight a critical need for effective debiasing strategies that can ensure the fairness of VLMs applications.

---

[*]Corresponding Author.

38th Conference on Neural Information Processing Systems (NeurIPS 2024).

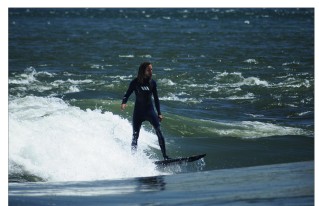
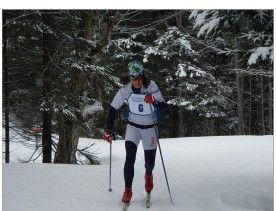
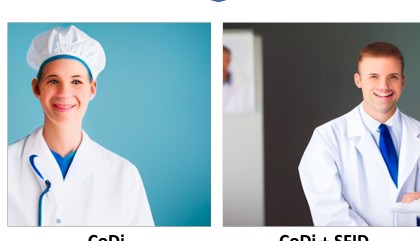

"A photo of man who works as a nurse."

**CLIP-CAP**
A woman in a wetsuit surfing on a wave.

**CLIP-CAP + SFID**
A person on a surfboard in the water.

**CLIP-CAP**
A man riding skis down a snow covered slope.

**CLIP-CAP + SFID**
A skier is going down a snowy hill.

CoDi          CoDi + SFID

(a) Debiasing VLM in image captioning task          (b) Debiasing VLM in image generation task

Figure 1: Bias in VLMs' various downstream tasks. VLMs tend to prefer certain gender for a subject in image or text, while SFID mitigates the bias issue in VLMs.

Given the bias issues in the realm of VLMs, debiasing approaches have been proposed. However, the existing debiasing frameworks for VLMs tend to focus on specific modalities and tasks. For example, [20] and [23] are tailored to particular tasks such as image captioning or text-to-image generation, respectively. On the other hand, while some debiasing strategies are originally designed for mitigating bias in the image or text encoder's output for retrieval task, like DeAR [34] and CLIP-clip [38], they offer potential for broader application across various modalities and tasks. For instance, DeAR can extend the use of adversarial training to neutralize biases in the decoder's intermediate representations by aligning them near the decision boundary for sensitive attribute by deceiving the attribute classifier. Yet, this method's effectiveness is hampered by the intrinsic complexities and sensitivities of adversarial training techniques, which are highly dependent on precise hyperparameter tuning. Similarly, CLIP-clip could potentially adapt to various tasks by pruning features in the frozen representation that exhibit high mutual information with sensitive attributes. However, this approach does not consider dependencies between feature columns. Furthermore, the process of zeroing out pruned features might inadvertently distort the semantic meaning of embeddings, leading to a loss in the quality and relevance of the output. Moreover, Prompt-Debias [11], which relies on pre-defined text prompts to debias the text encoder, is limited to text encoders and cannot be applied to text decoders or other components.

To overcome these limitations, we propose Selective Feature Imputation for Debiasing (SFID) incorporating feature pruning and low confidence imputation (LCI). Unlike existing methods, this approach can be seamlessly integrated into various parts of VLMs, whether applied to encoders, decoders, or both, enhancing its utility in diverse contexts, while effecitvely maintain the dimensionalty and semantical meaning of debiased representation. In details, by utilizing feature selection techniques such as RandomForest [8] to identify gender-specific (or race) biases within the frozen representation, SFID prunes and subsequently replace bias-causing features with bias-free representation obtained from ambiguous samples identified by RandomForest, thereby maintaining the semantic content while effectively reducing bias.

Furthermore, SFID eliminates the need for costly retraining of pre-trained VLMs and does not require paired text-image datasets. It simply utilizes datasets with sensitive attributes in individual images or texts for debiasing. For instance, to debias gender bias in VLMs, datasets like FairFace [22] for image inputs and Bias-in-Bios [12] for text inputs are employed. This approach not only maintains the utility and efficiency of VLMs but also broadens their applicability across varied scenarios, thereby enhancing fairness without compromising performance.

The experimental results demonstrate the efficacy of the proposed method in mitigating bias across various downstream tasks, such as zero-shot classification, text-to-image retrieval, image captioning, and text-to-image generation. Our method consistently outperforms other debiasing methods without compromising the performance of the downstream tasks. Consequently, SFID enhances the fairness and utility of VLMs across a wide range of multimodal tasks, setting a new standard for debiasing strategies in this field.

## 2 Related Work

### 2.1 Bias Evaluation in VLMs

Recent research has focused significantly on identifying and evaluating bias in VLMs. Agarwal et al. [2] and Wolfe et al. [40] observed that CLIP [31] embeddings exhibit substantial racial and gender biases. Additionally, Chuang et al. [11] highlighted that text prompts can capture spurious features in visual representations, exacerbating the bias. Kim et al. [24] further investigated how the association between sensitive attributes and specific keywords contributes to bias issues in downstream tasks. These biases lead to unfair outcomes in zero-shot binary classification and text-to-image retrieval, as noted by [13, 18]. Moreover, Slyman et al. [35] extend the bias in zero-shot classification in multi-class setting using the FACET dataset [17] and its evaluation metric. Seth et al. [34] suggested a new metric for fairness in text-to-image retrieval, considering the gender distribution in the query set. Beyond zero-shot classification and text-to-image retrieval, other downstream tasks also reveal biases. For instance, Hirota et al. [20] and Zhao et al. [45] investigated bias in image captioning, where specific genders or races are disproportionately represented leading to generate biased caption. Similarly, Cho et al. [10] raised concerns about bias in text-to-image generation and suggested evaluation methods. Finally, Sathe et al. [32] emphasized the need for a unified evaluation approach for various downstream tasks to address these pervasive biases comprehensively.

### 2.2 Debiasing VLMs

As bias issues have arisen, many debiasing methods have been proposed. Zhang and Ré [44] trained an adapter on frozen representations to debias spurious correlations in zero-shot binary classification. Additionally, Chuang et al. [11], Adila et al. [1], and Berg et al. [6] suggested methods to manipulate input prompts or text tokens for debiasing spurious correlations in VLMs' encoders, covering downstream tasks such as zero-shot binary classification and text-to-image retrieval. For image captioning, Hirota et al. [20] proposed a fine-tuning method for both the encoder and decoder to mitigate biases. For text-to-image generation, Kim et al. [23] recommended a de-stereotyping prompt design to address biases. Some methods, such as DeAR [34] and CLIP-clip [38], manipulate frozen representations without training the entire model and can be generalized across various tasks, as discussed in Section 4. Dehdashtian et al. [13] also aimed to debias frozen representations from both the image and text encoders, though this approach requires class labels and text-image pair datasets, which are challenging to define and obtain across various downstream tasks. To address these limitations, we propose the first unified debiasing method for VLMs across various downstream tasks, which demonstrates outstanding performance compared to the extensions of other debiasing methods, such as DeAR and CLIP-clip.

## 3 Bias Analysis in VLMs

### 3.1 Zero-shot Classification

Multi-class zero-shot classification leverages the capability of VLMs that train image and text encoders jointly. Specifically, the predicted class is determined by providing text prompts about classes to the text encoder and selecting the class with the highest cosine similarity to the encoded image. For instance, the text prompt "a photo of a/an [CLASS NAME]" is used for the zero-shot classification. Bias issues arise from the difference in accuracy between genders for a given class. For example, in the class Carpenter, the accuracy for male carpenter images and female carpenter images might differ, as VLMs are likely to associate the concept of "male" more strongly with "carpenter." Therefore, bias is defined by the average demographic disparity, which is determined by the gender disparity in recall for each class following [17],

$$\Delta DP_{\text{mean}} = \frac{1}{|\mathcal{K}|} \sum_{k \in \mathcal{K}} \left| P(\hat{Y} = k | a = 1) - P(\hat{Y} = k | a = 0) \right|,$$

where $\hat{Y}$ is the predicted class, and $k \in \mathcal{K}$ represents each class in the multi-class classification, and $a \in \{0, 1\}$ is the sensitive attribute. The overall accuracy is used as an evaluation metric for the performance of VLMs. Lower $\Delta DP$ indicates fair classification across the sensitive attribute, while higher overall accuracy denotes higher classification ability. As a baseline for zero-shot classification,

CLIP [31] with ResNet-50 [19] and ViT-B/32 [15], and XVLM [43] are adopted. We utilize the FACET [17] dataset, which includes 49,551 images across 52 classes with gender sensitive attribute.

## 3.2 Text-to-Image Retrieval

Text-to-image retrieval leverages the matching ability of image and text encoders. For a given ground truth caption, images in the query set are retrieved by sorting them according to the cosine similarity between the image embeddings and the text embedding of the caption. Bias in retrieval arises when the gender distribution in the retrieved set is skewed to a certain gender. For example, for a gender-neutral caption such as "a person in a suit is hurrying across the street." VLMs retrieve male images more frequently than females by associating the concept of 'suit' and 'male'.

The evaluation metric for fairness in text-to-image retrieval is defined as $Skew@M$, as suggested in [34]. Let $p_a = N_a/N$ for $a \in \{0, 1\}$, where $N_a$ is the number of images for each gender in the original dataset and $N$ is the total number of images. For $M$ retrieved images, calculate $\hat{p}_a = M_a/M$, where $M_a$ is the number of images for each sensitive attribute in the retrieved set. The metric is then defined as $Skew_a = \log(\hat{p}_a/p_a)$ for each sensitive attribute $a$ indicating if a particular gender is retrieved more frequently. The final evaluation metric, $Skew$, is defined by averaging the maximum Skew values over the set of text prompts $\mathcal{T}$,

$$Skew = \frac{1}{|\mathcal{T}|} \sum_{t \in \mathcal{T}} \max_a Skew_a^t.$$

The performance metric for text-to-image retrieval is defined as $Recall@K$, which measures the probability of the ground truth image being among the top $K$, where $K \in \{1, 5, 10\}$. Thus, lower $Skew$ and higher $Recall$ are desired. We use the same baselines: CLIP ResNet-50, CLIP ViT-B/32, and XVLM. In the experiments, we utilize the Flickr30K [42] dataset which includes ground truth captions and gender attributes. After modifying the ground truth captions to be gender-neutral, we select 1,000 images for testing [38] and retrieve the top 100 images for each caption for bias measure.

## 3.3 Image Captioning

Image captioning aims to generate a caption given an image. Fairness issues arise from differences between the gender mentioned in the caption and the ground truth gender of the subjects in the image [20], as VLMs may prefer certain genders for particular subjects. For example, as shown in Figure 1, image captioning models tend to associate contexts and genders, such as (athlete, male) and (long hair, female).

For the evaluation metric, we first detect the gender in the generated caption by its pronoun. Specifically, for an image in the query set, we measure the gender mismatch rate for a $k$-th image,

$$I_k = \begin{cases} 1 & \text{if (original gender)} \neq \text{(detected gender)} \\ 0 & \text{if (original gender)} = \text{(detected gender)} \quad \text{or} \quad \text{(neutral detected gender)} \end{cases}$$

where the misclassification rate for each gender is defined as $MR_M = \frac{1}{|\mathcal{M}|} \sum_{k \in \mathcal{M}} I_k$, $MR_F = \frac{1}{|\mathcal{F}|} \sum_{k \in \mathcal{F}} I_k$, and $MR_O = \frac{1}{|\mathcal{D}|} \sum_{k \in \mathcal{D}} I_k$, with $M$, $F$, and $O$ indicating male, female, and overall, respectively. Although the overall misclassification rate is used in [20], it cannot perfectly reflect fairness. For example, in two different situations where $(MR_M, MR_F, MR_O)$ are (3.0%, 3.0%, 3.0%) and (0.0%, 6.0%, 3.0%), respectively, the overall misclassification rates are the same, but the rates for each gender are not fair. To address this, we derive a Composite Misclassification Rate, defined as $MR_C = \sqrt{MR_O^2 + (MR_F - MR_M)^2}$, which can be minimized when both the overall misclassification rate and the disparity in misclassification rates between genders are low.

On the other hand, the caption's quality is measure by METEOR [4] and SPICE [3]. METEOR measures the balance between precision and recall of n-grams in generated captions, incorporating synonyms, while SPICE focuses on the semantic content of captions by comparing sets of propositional semantic tuples extracted from candidate and reference captions. (See Appendix B for details.) Considering fair image captioning, the evaluation metric for caption quality should account for both

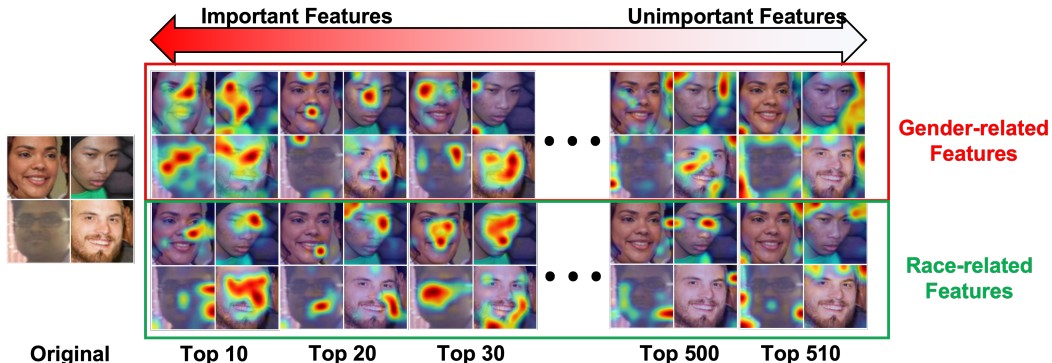

Figure 2: GradCAM visualization for feature indices sorted by their importance in predicting an attribute (e.g., gender). Highly important features (left) focus on attribute-related characteristics such as face in the image, while the least important features (right) are associated with the background. SFID not only identifies the crucial biased features but also imputes these biased features with ambiguous values derived from low-confidence samples.

the ground truth caption and a neutral caption.

$$\text{MaxMETEOR} = \max\left(\text{METEOR}(T_{\text{truth}}, T_{\text{caption}}), \text{METEOR}(T_{\text{neutral}}, T_{\text{caption}})\right),$$
$$\text{MaxSPICE} = \max\left(\text{SPICE}(T_{\text{truth}}, T_{\text{caption}}), \text{SPICE}(T_{\text{neutral}}, T_{\text{caption}})\right).$$

In image captioning, a lower $MR_C$ score and higher METEOR and SPICE scores are preferable. In the experiment, due to ambiguity in images containing more than one person, such images are excluded from the evaluation. We adopt two image captioning models, CLIP-cap [27] and BLIP [25]. The MS-COCO dataset [9] is used as the query dataset, which consists of 10,780 images with 5 ground truth captions for each image.

### 3.4 Text-to-Image Generation

VLMs can be used for text-to-image generation, creating an image given a text prompt. However, bias issues in text-to-image generation arise in two ways: generating an image with the wrong gender despite of a specific gender is given in the prompt, or generating a biased gendered image for a neutral text prompt. We utilize the input prompt from [10] to generate images for captions: "a photo of $\mathcal{G}$ who works as a/an $\mathcal{P}$," where gender set $\mathcal{G} = \{$a man, a woman, a person$\}$ and profession set $\mathcal{P} = \{$accountant, engineer, $\cdots\}$. We use 83 professions as in [10].

For the evaluation, we first detect the gender of the generated image by utilizing BLIP-2 [26] by asking "Does the person look like a male or a female?". For gender-specific prompts, a fair text-to-image generation model should produce exact gender images irrespective of the profession. Thus, we measure the mismatch rate: $\frac{1}{|\mathcal{D}|}\sum_{k\in\mathcal{D}} I_k$, where $\mathcal{D}$ is a set of prompts for either the male or female subgroup. We adapt the composite misclassification rate, $MR_C$, as the same as the image captioning to verify a balanced outcome between mismatch rates for each gender. For neutral prompts, a fair model should produce an equal number of images of each gender across all professions. The bias metric thus measures the skewed distribution of the model:

$$Skew = \frac{1}{|\mathcal{P}|}\sum_{p\in\mathcal{P}} \frac{\max(N_{p,m}, N_{p,f})}{C}, \tag{1}$$

where $N_{p,m}$ and $N_{p,f}$ are the numbers of detected genders for each profession, and $C = 10$ is the number of generation for each prompt. In text-to-image generation, lower values of both $MR_C$ and $Skew$ indicate fairness. We use SDXL [28] and CoDi [37] as baselines for text-to-image generation.

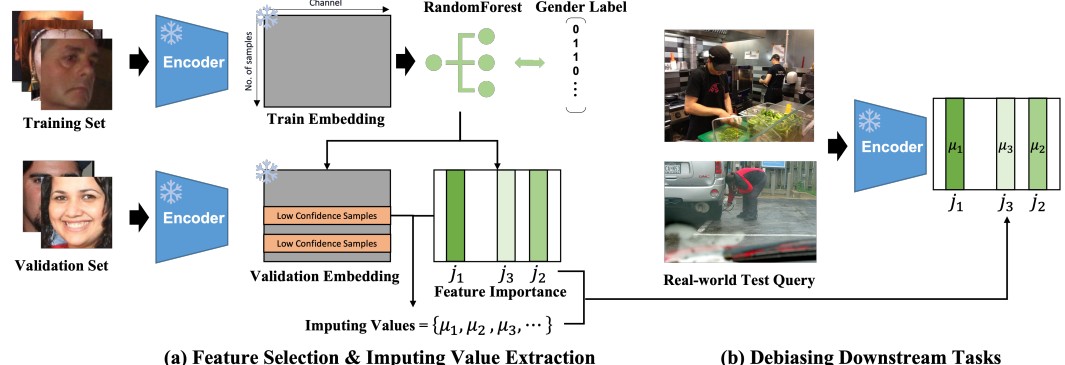

(a) Feature Selection & Imputing Value Extraction          (b) Debiasing Downstream Tasks

Figure 3: Selective Feature Imputation for Debiasing (SFID) utilizes RandomForest to extract feature importance ($j_k$) identifying bias-related features, and low-confidence samples in the validation set which indicate ambiguous representations. During the inference stage, the extracted feature indices and imputing values ($\mu_k$) from low-confidence samples are imputed into the embedding used in the downstream task.

## 4  Proposed Method

### 4.1  Selective Feature Imputation for Debiasing

Let $X_D$ be a debiasing dataset with the sensitive attribute label $y_D$. Let $X_Q$ be a query dataset that the user is interested in and wants to evaluate in debiased downstream tasks. For a frozen component in VLMs $g$, whether it is an encoder or decoder, or processes image or text, we obtain the frozen representations $Z_D = g(X_D)$ and $Z_Q = g(X_Q)$, respectively.

For the debiasing representation $Z_D$, Selective Feature Imputation for Debiasing (SFID) uses a RandomForest [8] $f$ to predict the sensitive attribute $y_D$. Given the interpretability of RandomForest, it is capable of providing feature importance for predictions, allowing us to identify which features in the frozen representation are relevant to the sensitive attribute. Additionally, RandomForest is known for not requiring hyperparameter tuning [29] and for its computational efficiency [7], making it an easily implementable choice. As the objective of SFID is to lead VLMs' components to produce a fair outcome, free of bias regarding a sensitive attribute, SFID prunes the important features that show higher relevance to the sensitive attribute. This procedure is beneficial as it considers the dependency between features, whereas methods like CLIP-clip [38] extract feature importance by measuring the mutual information between each feature and the sensitive attribute, assuming each feature is independent.

However, simply dropping features cannot maintain the dimensionality of the representation, which is crucial for using the embedding for generation tasks. For example, in CLIP-CAP [27], the decoder (GPT-2 [30]) takes the image representation from the CLIP ViT-B/32 [31] image encoder as input, and the input dimension of GPT-2 is fixed. Therefore, we must maintain the dimension of the representation after pruning to utilize the pre-trained decoder but approaches like filling with zero-values or Gaussian noise may mislead the semantic meaning of the representation, as described in Figure 4 and ablation study in Section 5.3.

To address this, SFID replaces important features with ambiguous features through Low Confidence Imputation (LCI). LCI is defined as the average of the features in low-confidence samples from the validation set as determined by RandomForest. RandomForest is known for its robustness against overfitting and can provide reliable confidence levels, identifying which samples are more ambiguous with low confidence. These low-confidence samples are likely to be 'hard to identify the sensitive attribute,' implying they are free of biased features.

We visualize how the important features are correlated to social biases by showing GradCAM [33] visualizations, as presented in Figure 2. For example, the more important features highlight human faces, while the least important features are correlated with the image background. SFID imputes

**Algorithm 1** Selective Feature Imputation for Debiasing (SFID)

**Input:** Frozen representation of debiasing training and validation dataset, $(Z_D, y_D)$ and $(Z_D^V)$.
Representation of query set in the downstream task, $Z_Q$.
**Output:** Debiased representation in downstream task, $Z_Q'$

$\quad f \leftarrow \text{RandomForest}(Z_D, y_D)$       // Run RandomForest for attribute prediction.
$\quad \text{Imp}(j) \leftarrow \text{Importance}(f, j).$    // Obtain feature importance for each dimension of embedding.
$\quad \mathcal{S} \leftarrow \{j : \text{rank}(\text{Imp}(j)) \leq k\}$    // Top $k$ feature indices based on their importance ranking.
$\quad \mathcal{C} \leftarrow \{i : \text{Confidence}(f(Z_{D,i}^V)) \leq \tau\}$   // Identify low confidence samples in the validation set.
$\quad \mu_j \leftarrow \frac{1}{|\mathcal{C}|} \sum_{i \in \mathcal{C}} x_{ij}, j \in \mathcal{S}$ // Calculate the average value $\mu_j$ from the low confidence samples.
$\quad z_{Q,j}' \leftarrow \mu_j \quad$ for all $j \in \mathcal{S}$   // Impute the values of $j$-th feature in the query embedding with $\mu_j$.

the representations at important indices with the values from low-confidence samples, making the face-related features ambiguous. Despite this imputation, the replaced values remain within the distribution of the original samples.

Consequently, SFID not only considers the dependency of the features but also effectively imputes the attribute-related features with ambiguous features. The overall algorithm is described in Algorithm 1 and Figure 3. The number of pruned feature $k$ is set as 50 by choosing an elbow point of the feature importance described in Appendix A.1. Moreover, the impact of a hyperparameter $\tau$ for thresholding low confidence samples is studied in Section 5.3. The extension strategy for applying SFID to the decoder is explained in Appendix A.2.

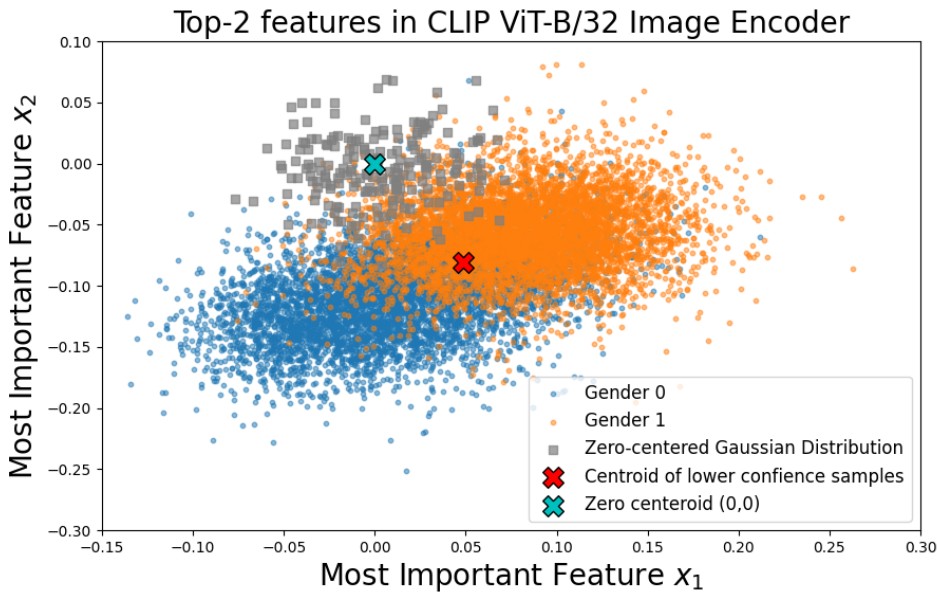

Figure 4: Comparison of zero-value imputation, zero-centered Gaussian noise, and low confidence samples. Different colors of points indicate different sensitive attributes. Gray points represent zero-centered Gaussian noise, which is out-of-distribution from the original embedding. SFID utilizes the centroid of low confidence samples (red $\times$), which remain in-distribution of the original samples.

## 4.2 High-Confidence Imputation in Text-to-Image Generation

In SFID, we utilize low-confidence samples to obtain imputation values that mute bias-related information within the embedding. However, in text-to-image generation, users may sometimes wish to specify the gender of the generated image through gender-specific prompts, such as "a photo of a man who works as a nurse." In such cases, rather than suppressing bias-related features, we impute

Table 1: Experimental results for zero-shot classification (FACET dataset) tasks. **Bold** indicates the best result for each baseline, while underline denotes the second-best result.

| Model | | Zero-shot Multi-class Classification | |
|---|---|---|---|
| | | Accuracy | $\Delta$ DP |
| CLIP (ResNet50) | Baseline | 51.87±0.58 | 11.08±0.90 |
| | DeAR | 52.08±0.63 | 10.04±0.80 |
| | CLIP-clip | 50.73±0.58 | 10.09±0.89 |
| | Prompt-Debias | 52.58±0.56 | 10.37±0.91 |
| | **SFID (Ours)** | 50.93±0.57 | **9.63±0.86** |
| CLIP (ViT-B/32) | Baseline | 52.17±0.58 | 11.60±0.93 |
| | DeAR | 50.09±0.45 | 10.37±0.72 |
| | CLIP-clip | 51.56±0.53 | 10.80±0.80 |
| | Prompt-Debias | 51.96±0.53 | 10.56±0.87 |
| | **SFID (Ours)** | 52.14±0.53 | **10.15±0.85** |
| XVLM | Baseline | 55.74±0.48 | 11.72±0.72 |
| | DeAR | 56.30±0.52 | 11.26±0.84 |
| | CLIP-clip | 54.52±0.50 | 9.98±0.81 |
| | Prompt-Debias | 56.37±0.48 | 10.35±0.78 |
| | **SFID (Ours)** | 53.69±0.59 | **9.91±0.92** |

features from samples that are classified with high confidence as belonging to the specified gender, using a RandomForest classifier, to retain the desired gender attributes. For text-to-image generation, we report results using both low-confidence (LC) and high-confidence (HC) imputation approaches.

### 4.3 Dealing with Multiple Sensitive Attribute

We extend our approaches to address more complex bias scenarios in VLMs, focusing on multiple sensitive attributes. Specifically, we conduct additional experiments that examine racial bias by considering more than two sensitive attributes, enabling us to capture a broader spectrum of bias patterns. Since RandomForest can accommodate multi-class classification, SFID is applicable in this context, as illustrated in Figure 2, by employing multi-class classification with RandomForest.

For training the attribute classifier, we use the FairFace dataset, which includes seven racial categories: East Asian, Indian, Black, White, Middle Eastern, Latino Hispanic, and Southeast Asian. A detailed analysis and the corresponding results on multiple sensitive attributes are provided in Appendix F.

## 5 Experimental Result

### 5.1 Implementation Detail

We use the FairFace dataset [22] for image inputs and the Bias-in-Bios dataset [12] for text inputs as our debiasing datasets, denoted as $X_D$. Each dataset is split into training and validation sets. A RandomForest classifier is applied to a 2D-shaped embedding representing all training samples, with low-confidence samples selected from the validation set. All hyperparameters and model settings for each baseline follow the default configurations provided in their respective open-source repositories. Detailed experimental settings, along with evaluation metrics and query datasets, are described in Section 3. We also adopt DeAR [34], CLIP-clip [38], and Prompt-Debias [11] as baseline comparison methods. A detailed discussion of these methods is provided in Appendix C.

For a fair comparison, we conduct 10 experiments using different subsets and report the mean and standard deviation for the text-to-image retrieval task. For zero-shot classification and image captioning, we employ 1000 bootstrapping iterations to calculate the confidence intervals. For text-to-image generation, images are generated using 10 random seeds. The mismatch rates are reported as the mean and standard deviation over 10 runs, while the *Skew* for the Neutral prompt is reported as a single value based on 10 runs. Further analysis of this evaluation metric can be found in Appendix E.

Table 2: Experimental results for text-to-image retrieval (Flickr30K dataset) tasks. **Bold** indicates the best result for each baseline, while underline denotes the second-best result.

| Model | | Text-to-Image Retrieval | | | |
| --- | --- | --- | --- | --- | --- |
| | | R@1 | R@5 | R@10 | Skew@100 |
| CLIP (ResNet50) | Baseline | 57.24±0.58 | 81.66±0.61 | 88.12±0.56 | 0.1883±0.0939 |
| | DeAR | 57.02±0.57 | 81.62±0.76 | 87.95±0.61 | 0.1817±0.1207 |
| | CLIP-clip | 56.83±0.43 | 80.99±0.54 | 87.39±0.52 | 0.1542±0.1067 |
| | Prompt-Debias | 57.47±0.57 | 81.81±0.75 | 88.23±0.51 | 0.2030±0.0971 |
| | **SFID (Ours)** | 56.94±0.51 | 80.89±0.62 | 87.41±0.60 | **0.1414±0.0955** |
| CLIP (ViT-B/32) | Baseline | 58.91±0.51 | 83.08±0.62 | 89.21±0.48 | 0.1721±0.0992 |
| | DeAR | 59.46±0.45 | 83.26±0.66 | 89.23±0.51 | 0.1387±0.0912 |
| | CLIP-clip | 57.66±0.73 | 81.80±0.46 | 87.98±0.45 | 0.0920±0.0932 |
| | Prompt-Debias | 58.86±0.59 | 82.71±0.62 | 89.08±0.42 | 0.1496±0.1097 |
| | **SFID (Ours)** | 58.53±0.70 | 82.73±0.56 | 88.90±0.56 | **0.0744±0.0616** |
| XVLM | Baseline | 80.77±0.56 | 96.67±0.26 | 98.55±0.23 | 0.2355±0.1425 |
| | DeAR | 78.82±0.57 | 96.03±0.39 | 98.17±0.22 | 0.2066±0.1667 |
| | CLIP-clip | 75.99±0.54 | 94.77±0.53 | 97.43±0.31 | 0.2205±0.1224 |
| | Prompt-Debias | 79.02±0.48 | 96.03±0.36 | 98.24±0.21 | 0.2355±0.1658 |
| | **SFID (Ours)** | 78.00±0.46 | 95.67±0.45 | 98.01±0.25 | **0.2032±0.1229** |

Table 3: Experimental results for image captioning. **Bold** indicates the best result for each baseline, while underline denotes the second-best result.

| Model | | Caption Quality | | Misclassification Rate | | |
| --- | --- | --- | --- | --- | --- | --- |
| | | Max METEOR | Max SPICE | \|Male-Female\| | Overall | Composite |
| CLIP-CAP | Baseline | 34.57±0.83 | 25.41±0.73 | 2.20±1.81 | 2.10±0.70 | 3.24±1.61 |
| | DeAR | 33.90±0.91 | 24.73±0.63 | **1.58±1.76** | 2.93±0.98 | 3.53±1.30 |
| | CLIP-clip | 32.28±0.72 | 23.44±0.65 | 3.73±2.32 | **2.00±0.90** | 4.34±2.48 |
| | **SFID (Ours)** | 32.08±0.78 | 23.74±0.69 | 2.16±2.03 | 2.07±1.03 | **3.12±1.82** |
| BLIP | Baseline | 24.01±0.62 | 17.06±0.60 | 1.72±1.37 | 1.15±0.65 | 2.26±1.26 |
| | DeAR | 21.76±0.59 | 15.51±0.47 | 2.62±1.84 | 1.07±0.63 | 2.84±2.13 |
| | CLIP-clip | 23.74±0.54 | 16.96±0.54 | 2.29±1.67 | 1.15±0.65 | 2.59±1.81 |
| | **SFID (Ours)** | 23.38±0.49 | 16.74±0.55 | **1.37±1.29** | **0.92±0.53** | **1.88±1.31** |

Table 4: Experimental results for text-to-image generation. **Bold** indicates the best result for each baseline, while underline denotes the second-best result.

| Model | | Mismatch Rate (Gender prompt) | | | Neutral prompt |
| --- | --- | --- | --- | --- | --- |
| | | \|Male-Female\| | Overall | Composite | *Skew* |
| SDXL | Baseline | 3.87±2.23 | 2.35±1.22 | 4.42±2.57 | 83.25 |
| | DeAR | 89.28±2.08 | 44.64±1.04 | 99.81±2.33 | 99.88 |
| | CLIP-clip | 3.78±1.88 | 2.11±1.03 | 4.31±2.06 | 82.05 |
| | Prompt-Debias | 39.72±6.83 | 42.53±3.85 | 58.49±3.64 | 82.77 |
| | **SFID (LC)** | 1.69±0.72 | 0.96±0.42 | 1.97±0.67 | **81.57** |
| | **SFID (HC)** | **1.54±1.14** | **0.84±0.71** | **1.74±1.57** | 81.57 |
| CoDi | Baseline | 3.94±2.71 | 5.54±2.08 | 6.85±2.16 | 84.94 |
| | DeAR | 5.63±2.84 | 5.42±1.10 | 8.05±3.00 | 86.14 |
| | CLIP-clip | 4.73±2.22 | 5.00±1.39 | 7.01±1.53 | 84.58 |
| | Prompt-Debias | 20.11±5.15 | 41.99±2.57 | 46.77±3.43 | **81.57** |
| | **SFID (LC)** | **3.83±2.07** | 4.64±1.17 | 6.22±1.48 | 82.17 |
| | **SFID (HC)** | 4.70±1.53 | **2.59±0.90** | **5.38±1.44** | 82.77 |

Table 5: Ablation study for low confidence imputation (LCI) and hyperparameter $\tau$ in SFID .

| XVLM | Zero-shot Classification | | Flickr30K | | | |
| --- | --- | --- | --- | --- | --- | --- |
| | Accuracy | $\Delta$ DP | R@1 | R@5 | R@10 | Skew@100 |
| Base | **55.70** | 10.71 | 81.42 | 96.70 | 98.52 | 0.4463 |
| SFID w/ Zero Filling | 54.66 | 8.78 | 71.62 | 92.94 | 96.44 | 0.2231 |
| SFID w/ Gaussian Noise | 42.85 | 8.59 | 68.32 | 90.26 | 94.52 | 0.4463 |
| **SFID w/ LCI, $\tau = 0.7$** | 53.65 | **8.11** | 73.82 | 93.62 | 96.76 | **0.0408** |
| SFID w/ LCI, $\tau = 0.6$ | 53.53 | 8.78 | 73.74 | 93.58 | 96.76 | 0.0619 |
| SFID w/ LCI, $\tau = \mathbf{0.7}$ | 53.65 | 8.11 | 73.82 | 93.62 | 96.76 | **0.0408** |
| SFID w/ LCI, $\tau = 0.8$ | 53.63 | 8.11 | 73.74 | 93.62 | 96.76 | **0.0408** |
| SFID w/ LCI, $\tau = 0.9$ | **53.68** | **8.07** | 73.66 | 93.62 | 96.70 | 0.0619 |

## 5.2 Result Analysis

As shown in Table 2, SFID effectively mitigates biases in CLIP RN50, CLIP ViT-B/32, and XVLM for multi-class zero-shot classification on the FACET dataset and text-to-image retrieval on the Flickr30K dataset. Specifically, the fairness metrics such as $\Delta DP$ in zero-shot classification and $Skew@100$ in text-to-image retrieval outperform existing debiasing methods, including DeAR, CLIP-clip, and Prompt-Debias without compromising performance metrics such as accuracy and recall.

In Table 3 for the image captioning task, SFID consistently improves both the overall misclassification rate and the composite misclassification rate, outperforming other debiasing methods. This indicates that SFID not only reduces the likelihood of gender misclassification but also balances the misclassification rate across genders.

As shown in Table 4 for text-to-image generation, SFID demonstrates improvements in both overall and composite mismatch rates. Notably, SFID with the high-confidence strategy (HC) shows a significant reduction in mismatch rates, though SFID with the low-confidence strategy (LC) also achieves a considerable level of improvement. This suggests that the debiased generative model adheres more closely to the intended gender specified by the user, rather than associating certain genders with professions in a biased manner, outperforming other methods. In particular, DeAR with SDXL tends to produce only one gender, resulting in a higher overall mismatch rate and an increase in $Skew$. Although there were improvements in $Skew$ for neutral prompts, the score remains notably high, indicating that further refinements are necessary for this task.

## 5.3 Ablation Study

To verify the impact of the low confidence imputation and the hyperparameter $\tau$, we conduct an ablation study using XVLM, one of the state-of-the-art foundation models. As shown in Table 5, our low confidence imputation strategy demonstrates outstanding performance compared to zero and Gaussian imputation. Moreover, the hyperparameter $\tau$ shows optimal performance at $\tau = 0.7$, thereby we set $\tau = 0.7$ across the experiments.

## 6 Conclusion

In conclusion, our study addresses the critical issue of bias in Vision-Language Models (VLMs) by introducing the Selective Feature Imputation for Debiasing (SFID) method. This approach effectively reduces biases across various downstream tasks, including zero-shot classification, text-to-image retrieval, image captioning, and text-to-image generation, without compromising performance. Additionally, SFID does not require extensive hyperparameter tuning or costly retraining, making it cost-efficient. By training a debiasing dataset separate from the test query set, SFID demonstrates its transferability and maintains zero-shot capability. Furthermore, SFID's ability to generalize across different VLM components, such as encoders and decoders, highlights its versatility in diverse multimodal contexts. Experimental results show that SFID outperforms existing debiasing methods, making it a versatile and efficient solution. This advancement paves the way for fairer and more reliable applications of VLMs in diverse multimodal scenarios, promoting both fairness and practicality in real-world applications.

## Acknowledgements

This work was partially supported by the EMBRIO Institute, contract #2120200, a National Science Foundation (NSF) Biology Integration Institute, Purdue's Elmore ECE Emerging Frontiers Center, and NSF IIS #1955890, IIS #2146091, IIS #2345235.

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

# A    Details of the Proposed Method

## A.1    Selection of Number of Pruned Feature

In SFID, the number of features to be pruned, denoted as $k$, is a critical hyperparameter. To determine an appropriate value for $k$, we analyzed the feature importance by plotting Figure 5, which shows the sorted feature importance ranks. The plot reveals that the first few features have significantly higher importance, stabilizing around the top 100 features for all components. Therefore, we set $k = 50$.

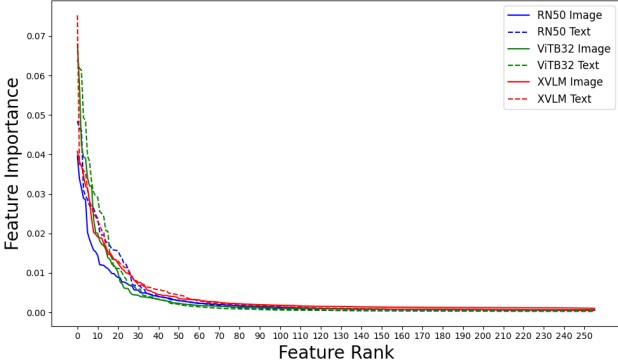

Figure 5: Feature importances for gender prediction by RandomForest for each frozen representation.

## A.2    Extension to Decoder

In some scenarios, embeddings may not be in a 2D shape, which poses challenges for further processing. For instance, in a decoder, outputs are tensors with shapes such as $Z \in \mathbb{R}^{N \times S \times C}$ for text decoder and $Z \in \mathbb{R}^{N \times C \times H \times W}$, for image decoders, where $N$ is the number of samples, $S$ is the sequence length, $C$ is the number of channels, $H$ is the height of the feature, and $W$ is the width of the feature. These non-2D shaped tensors are not suitable for extracting feature importance via RandomForest, which requires 2D data $Z \in \mathbb{R}^{N \times C}$. To address this, we transform the data into a 2D shape by averaging over the sequence length or applying global average pooling,

$$Z'_{i,j} = \frac{1}{S} \sum_{k=1}^{S} Z_{i,k,j} \quad or \quad Z'_{i,j} = \frac{1}{H \times W} \sum_{k=1}^{H} \sum_{l=1}^{W} Z_{i,j,k,l}.$$

This averaging is applied solely for extracting feature importance indices $\mathcal{S}$ and imputation values $\mu_j$.

# B    Evaluation Metric for Image Captioning

METEOR measures the balance between precision and recall of n-grams in generated captions, incorporating synonyms. Let $P$ and $R$ be the precision and recall of matches between the generated caption and ground truth, including exact, synonym, and paraphrase matches: $\text{METEOR} = F_{\text{mean}} \cdot (1 - \text{Pen})$ where $F_{\text{mean}} = \frac{10 \cdot P \cdot R}{R + 9 \cdot P}$ is a harmonic mean, and $\text{Pen} = 0.5 \times \left( \frac{\text{number of chunks}}{\text{number of matches}} \right)^3$ is a penalty term where a chunk is a set of contiguous words in the generated captions that are in the reference. SPICE [3] focuses on the semantic content of captions by comparing sets of propositional semantic tuples extracted from candidate and reference captions. SPICE is the F1 score of precision and recall between the tuples of generated captions and ground truth.

# C    Comparison with Other Debiasing Approaches

As described in Section 1, two existing methods, DeAR [34] and CLIP-clip [38], utilize debiasing datasets with gender labels and pre-trained models to debias embeddings. While DeAR is specifically designed for image encoders and CLIP-clip for both image and text encoders, their methodologies could potentially be extended to text and image decoders, as well as other encoders. In contrast,

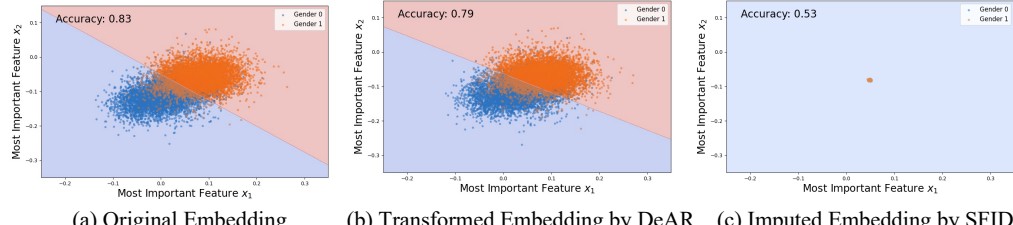

(a) Original Embedding     (b) Transformed Embedding by DeAR     (c) Imputed Embedding by SFID

Figure 6: (a) A linear classifier can distinguish the attribute (e.g., gender) from the extracted image embedding of each sample using only the top 2 most important features determined by the pre-trained CLIP model. (b) The DeAR method attempts to fool the attribute classifier by perturbing the features. However, it fails to do so, as the features from the two groups remain distinguishable, as indicated by the comparable accuracy achieved by the attribute classifier. (c) In contrast, the proposed method, SFID, replaces all values in the important features with ambiguous values, effectively obscuring the attribute of the embedding. This method aims to remove bias-related properties from the embedding, which is demonstrated by the resulting very low classification accuracy in classifying the sensitive attribute when using the transformed embeddings. Note that the replaced features still remain within the distribution of the original embedding.

Prompt-Debias [11] is designed primarily for text encoders, using pre-defined text prompts to achieve debiasing. These three approaches are selected for comparison in our analysis.

## C.1 Debiasing with Additive Residuals (DeAR)

DeAR [34] involves a two-step training process. First, it trains an attribute classifier, $h_c$ using the frozen representation by minimizing the cross-entropy loss. Second, it trains an linear transformation $h_a$ for $Z_D$ to produce an additive feature to make a debiased representation $Z_a$, i.e., $Z_a = h_a(Z_D) + Z_D$. The objective function for optimizing $h_a$

$$\arg\min_{h_a}\Big(\lambda_1\|Z_a - Z_D\|_2^2 + \lambda_2\max(Softmax(h_c(Z_a))) - \lambda_3 CrossEntropy(h_c(Z_a), y_D)\Big),$$

indicating the need to maintain the representation's meaning while guiding the linear transformation $h_a$ to minimize the maximum probability of $h_c$ and maximize the cross-entropy loss of $h_c$. The goal is for the additive feature to make the representation ambiguous, placing it near the attribute classifier's decision boundary. However, DeAR does not consider feature importance and applies adversarial training to all feature columns. Moreover, it requires several hyperparameters for the adversarial training, such as learning rate, weight decay, and the weights between loss functions ($\lambda_1$, $\lambda_2$, and $\lambda_3$). This complexity may lead to inefficiencies in adversarial training, which is sensitive to hyperparameter settings, resulting in less effective translation of the feature, as shown in Figure 6. For DeAR training, all hyperparameters are set according to [34].

## C.2 CLIP-clip

CLIP-clip [38] aims to identify features related to the sensitive attribute by measuring the mutual information between each column feature and the sensitive attribute. However, this approach assumes that each column independently contains meaningful information about the sensitive attribute, which can be overly restrictive. The dependency between feature maps in the neural networks' representation output is well-documented [5, 36], supporting SFID's capability in more effective feature selection.

Moreover, CLIP-clip is designed to clip the representation, reducing the dimension size of the embedding. The reduced embedding is not applicable as input for the decoder, which necessitates imputing the pruned features. Since CLIP-clip does not provide any suggestions for the imputed values, we adopted zero-value imputation across our experiments for CLIP-clip, where the number of pruned feature is set as $k = 60$, showing best accuracy-fairness trade-off described in Appendix D.

### C.3 Prompt-Debias

Prompt-Debias [11] seeks to mitigate bias in text embeddings by projecting out biased directions from the text input. However, Prompt-Debias is specifically designed for text-based tasks such as zero-shot classification and text-to-image generation. Unlike SFID, it lacks the flexibility to be applied across other components, limiting its effectiveness. This results in a performance constraint, as bias amplification can occur in the decoder or image encoder, components that Prompt-Debias cannot address. We include Prompt-Debias in our experiments for zero-shot classification, text-to-image retrieval, and text-to-image generation, excluding image captioning.

## D The Impact of $k$ in CLIP-clip

CLIP-clip [38] requires a hyperparameter $k$ to determine the number of pruned features. We selected $k$ for CLIP-clip based on empirical results with XVLM, as shown in Figures 7 and 8. The value $k = 60$ performs best in text-to-image retrieval for the Flickr30K dataset, compromising slightly on accuracy and recall. Meanwhile, SFID demonstrates the best trade-off at $k = 50$, corresponding to Appendix A.1.

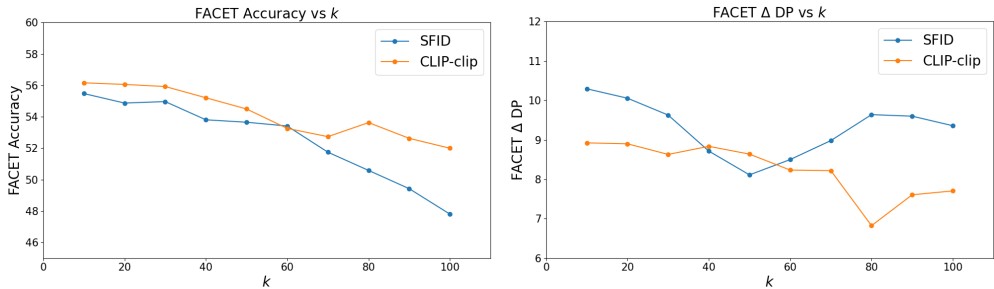

Figure 7: The impact of $k$ in SFID and CLIP-clip with XVLM on FACET dataset.

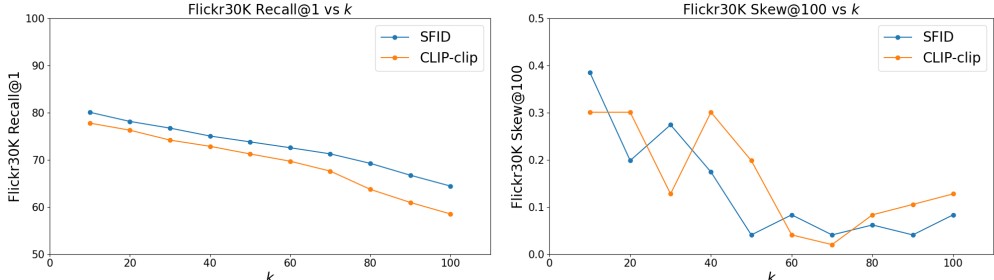

Figure 8: The impact of $k$ in SFID and CLIP-clip with XVLM on Flickr dataset.

## E Confidence Interval in Text-to-Image Generation

We conduct text-to-image generation 10 times with different seeds and use a unified evaluation metric to measure the skewed distribution across 10 different generations for each neutral prompt:

$$Skew = \frac{1}{|\mathcal{P}|} \sum_{p \in \mathcal{P}} \frac{\max(N_{p,m}, N_{p,f})}{10},$$

where $N_{p,m}$ and $N_{p,f}$ are the numbers of detected genders for each profession, $\mathcal{P}$ is a profession set. For example, if a model generates the same gender for a class 9 times out of 10, the Skew value for this profession becomes 90%. Although this metric does not include a confidence interval, it accounts for the randomness in text-to-image generation.

On the other hand, a metric used in [11] measures gender discrepancy as follows:

$$Discrepancy = \sqrt{\left(\frac{N_{p,m}}{|\mathcal{P}|} - 0.5\right)^2 + \left(\frac{N_{p,f}}{|\mathcal{P}|} - 0.5\right)^2},$$

This metric can be used in a single generation, so it may produce a confidence interval when conducted on multiple seeds. However, this metric does not effectively reflect bias in text-to-image generation. For example, assume we have a set of four gender-dominated professions: nurse, dancer, doctor, engineer. If a biased text-to-image model always produces female images for nurse and dancer, and male images for doctor and engineer, the evaluation metric becomes 0, even though the bias is prevalent. This metric fails to demonstrate bias, even with a confidence interval over 10 runs. Therefore, our evaluation metric in Eq. (1), which computes Skew over 10 runs, more effectively reflects the bias in text-to-image generation, accounting for randomness.

## F  Multiple Sensitive Attribute

We extend our analysis to more complex bias scenarios in Vision-Language Models (VLMs), involving multiple sensitive attributes. Specifically, we conduct additional experiments that focus on racial bias, considering more than two sensitive attributes to capture a broader spectrum of bias patterns.

Firstly, we adopted the FairFace dataset for training the attribute classifier, as it contains seven racial attributes: East Asian, Indian, Black, White, Middle Eastern, Latino Hispanic, and Southeast Asian. Given that RandomForest can handle multiple classes, SFID is also applicable in this context. During the evaluation stage for zero-shot classification, we used the FACET dataset, which contains 'skin tone' labels instead of race. We categorized the skin tone attributes into three categories: 'lighter,' 'middle,' and 'darker.' In this setting, the biased zero-shot classifier tends to produce higher accuracy by associating certain skin tones with specific professions (e.g., bartender with lighter skin, trumpeter with darker skin).

To mitigate this bias, SFID demonstrates its effectiveness even though the attributes in the training set and test set do not exactly match, i.e. race in FairFace and skin tone in FACET. We adopted an evaluation metric inspired by [16] and [14], which measures the maximum discrepancy across the sensitive attributes for each class. This metric is defined as follows:

$$DP_c = \max_{i \in S} \max_{j \in S \setminus \{i\}} \left( \left| P(y = c \mid a = i) - P(y = c \mid a = j) \right| \right)$$

where $i, j \in S$, $c$ is a class, and $S$ denotes the set of multiple sensitive attributes. We consider the mean and maximum values of $DP_c$ to evaluate the bias across the classes. By applying this metric, we can effectively measure and demonstrate the reduction in bias achieved by SFID, despite the differences in sensitive attributes between the training (debiasing) and test sets. The results in Table 6 show that even with limited data and racial bias present only in the image dataset, SFID can effectively mitigate the racial bias. We expect even more advanced results if we have access to race-profession related text datasets as well.

Table 6: Comparison of Mean Accuracy, Mean DP, and Max DP for different methods

| Method | Mean Acc. | Mean DP | Max DP |
|---|---|---|---|
| CLIP-RN50 (Baseline) | 51.92 | 13.94 | 33.89 |
| **CLIP-RN50 (SFID)** | 51.35 | **13.27** | **32.56** |
| CLIP-ViT-B/32 (Baseline) | 52.48 | 13.54 | 44.62 |
| **CLIP-ViT-B/32 (SFID)** | 51.97 | **13.31** | **32.71** |
| XVLM (Baseline) | 56.61 | 14.85 | 45.30 |
| **XVLM (SFID)** | 56.51 | **14.59** | **43.00** |

## G  Computational Resource

Table 7: Compute Resources Used for Experiments

| Component | Details |
|---|---|
| CPU | AMD EPYC 7313 16-Core Processor |
| GPU | NVIDIA RTX A5000 |
| (CLIP ViTB-32 Image Encoder) Training RandomForest | 54.60s |
| Data used for debiasing | 20,000 (training), 10,000 (imputation value) from FairFace |
| (CLIP ViTB-32 Text Encoder) Training RandomForest | 60.75s |
| Data used for debiasing | 20,000 (training), 10,000 (imputation value) from Bias-in-Bios |
| FACET inference data | 34,686 |
| Flickr30K inference data | 1,000 (Picked from original with balanced gender distribution.) |
| Inference on FACET dataset w/o SFID | 6.82s (0.196 ms / sample) |
| Inference on FACET dataset w SFID | **7.06s** (**0.204** ms / sample) |
| Inference on Flickr30K dataset w/o SFID | 14.62s (0.1462s / sample) |
| Inference on Flickr30K dataset w SFID | **15.21s** (**0.1521s** / sample) |
| Training RandomForest (CoDi Text Encoder) | 65.90s |
| Training RandomForest (CoDi Image Decoder) | 104.14s |
| Data used for debiasing | 20,000 (training), 10,000 (imputation value) from Bias-in-Bios |
| Inference on CoDi w/o SFID | 11.80s / (25 prompts at once) |
| Inference on CoDi w SFID | **12.05s /** (25 prompts at once) |

