# OpenReview forum: "A Unified Debiasing Approach for Vision-Language Models across Modalities and Tasks"
_NeurIPS.cc/2024/Conference — NeurIPS 2024 spotlight_

### Official Review · Reviewer_Wrj8 · 2024-07-12

**Soundness:** 3
**Presentation:** 3
**Contribution:** 2
**Rating:** 6
**Confidence:** 4

**Summary:**

The authors propose a method for debiasing the representations of Vision-Language Models (VLMs) that can be applied at various layers of the image and text encoders/decoders, and can be used for a variety of downstream tasks such as image generation, 0-shot classification, text to image retrieval, and image captioning.

**Strengths:**

- This work's goal of debiasing VLMs is an important and timely problem that should be of interest to many in the NeurIPS community
- The proposed approach is seemingly effective, lightweight and seems easy to apply
- The method is finetuning-free, which mitigates issues such as catastrophic forgetting

**Weaknesses:**

**Minor issues:**

- The writing could be improved in parts. There are minor grammatical errors in parts, such as "For instance, Hirota et al. [18], Zhao et al. [42] investigated bias in image captioning, where specific genders or races are disproportionately represented **leading to generate** biased caption." I suggest another editing pass or two.
- I find the structure of section 3 "Bias Analysis in VLMs" is a little odd. It mostly reads like background/preliminary material, but it also contains some experimental setup details (such as in line 165). This seems off, as this is before the proposed method section. I suggest moving these experimental details to section 5, with the rest of the experimental setup.
- Figure 4 shows that DeAR fails to produce bias-free representations. Including a figure likes this naturally makes the reader want to see a figure showing that the proposed approach **does** succeed here. It feels incomplete as-is.

**More significant issues:**

- The experimental results don't include confidence intervals. The authors justify this by pointing out that the proposed approach is deterministic. However, CIs still can and should be calculated by an approach such as bootstrapping the test set.
- Relatedly, the text to image generation experiment is **not** deterministic if the text2img generator's random seed is changed, so CIs should definitely be reported here.
- The compared methods are limited to only 2 other debiasing approaches and the baseline model.
- The proposed approach of 1) finding features most associated with the biased attributes and 2) replacing these features with an imputed value from low-confidence samples is, I think, very similar in practice to Chuang et al.'s approach [1] in "Debiasing Vision-Language Models via Biased Prompts". Chuang modifies embeddings by projecting them onto the subspace defined by the linear direction most associated with the biased attribute(s). Intuitively, the direction most associated with the biased attribute should be similar to the direction defined by the features identified in step 1) of the proposed approach. Not that I am **not** saying that this similarity alone is a limitation; I think the proposed approach is different enough to still have sufficient novelty. However, I think that Chuang's approach really needs to be compared against (for all experiments except maybe image captioning), in order to verify that these features are either finding a different direction in the embedding space or that the low confidence imputation performs better than just projecting onto the bias subspace.

[1] Chuang, Ching-Yao, et al. "Debiasing vision-language models via biased prompts." arXiv preprint arXiv:2302.00070 (2023).

**Questions:**

- Is the improvement of the proposed approach over the compared methods significant when CI are considered, where these CI's are obtained from bootstrapping or some other way?
- How does the proposed approach perform when compared against the method proposed in "Debiasing vision-language models via biased prompts."?

**Limitations:**

I think the discussion on limitations could be improved. The authors point to Section ***5.2 Result Analysis** as their discussion of limitations. 5.2 covers the results of their experiments, and does include some discussion on how the experimental results on image generation point to room for improvement. However, I feel like a discussion of the limitations would be greatly improved by dedicating a separate section/subsection to it. I would expect the authors to discuss more fundamental limitations, rather than just experimental performance. For instance, I would include details on how the method assumes access to a validation set with labels for bias/protected attributes (e.g. a dataset with race/gender/etc labels).

---

> ### Author Rebuttal · Authors · 2024-08-06
>
> ### Confidence interval in Text-to-Image Generation
> Thank you for pointing out the lack of confidence intervals in our experimental results.
>
> We acknowledge that text-to-image generation is not deterministic. This is why we conduct text-to-image generation 10 times with different seeds and use a unified evaluation metric to measure the skewed distribution across 10 different generations for each neutral prompt:
> \begin{align*}
> Skew = \frac{1}{|\mathcal{P}|} \sum_{p \in \mathcal{P}} \frac{\max(N_{p,m}, N_{p,f})}{10},
> \end{align*}
> where $N_{p,m}$ and $N_{p,f}$ are the numbers of detected genders for each profession, $\mathcal{P}$ is a profession set. For example, if a model generates the same gender for a class 9 times out of 10, the Skew value for this profession becomes 90%. Although this metric does not include a confidence interval, it **accounts for the randomness in text-to-image generation**.
>
> On the other hand, a metric used in [1] measures gender discrepancy as follows:
> \begin{align*}
> Discrepancy=\sqrt{\Bigl(\frac{N_{p,m}}{\vert \mathcal{P}\vert}-0.5\Bigr)^2 + \Bigl(\frac{N_{p,f}}{\vert \mathcal{P}\vert}-0.5\Bigr)^2},
> \end{align*}
> This metric can be used in a single generation, so it may produce a confidence interval when conducted on multiple seeds. However, this metric does not effectively reflect bias in text-to-image generation. For example, assume we have a set of four gender-dominated professions: {nurse, dancer, doctor, engineer}. If a biased text-to-image model always produces female images for nurse and dancer, and male images for doctor and engineer, the evaluation metric becomes 0, even though the bias is prevalent. This metric fails to demonstrate bias, even with a confidence interval over 10 runs. Therefore, our evaluation metric, which computes Skew over 10 runs, more effectively reflects the bias in text-to-image generation, accounting for randomness.
>
> On the other hand, demonstrating a confidence interval for the mismatch rate for gender-prompt is possible as it can be measured by a single experiment. As we have already executed the experiment with 10 different seeds,  we report the confidence interval for mismatch rates in **Table 1 of the rebuttal PDF file**.
>
> [1] Chuang, Ching-Yao, et al. "Debiasing vision-language models via biased prompts." arXiv preprint arXiv:2302.00070 (2023).
> ### Confidence interval for all downstream tasks
> We agree with the importance of including confidence intervals in experimental results, even when the model is deterministic, as is the case with pre-trained models used in zero-shot classification, text-to-image retrieval, and image captioning.
>
> For zero-shot classification and image captioning, we conducted experiments on the full test dataset. We utilized the bootstrapping technique by generating 1000 datasets with replacement and reported the confidence intervals in **Table 1 of the rebuttal PDF file** for zero-shot classification and image captioning.
>
> In the case of text-to-image retrieval, the test set is a subset of the entire Flickr30K dataset, created by randomly sampling 1000 samples. We extended the experiments by using 10 different test sets to obtain the confidence intervals. These results are also included in **Table 1 of the rebuttal PDF file**.
>
> #### Robust superiority of SFID
> Overall, when we include confidence intervals for all downstream tasks via bootstrapping or additional experiments, SFID consistently outperforms the comparison methods in all cases. **This robust performance across various scenarios highlights the effectiveness and reliability of SFID in mitigating bias.**
>
> ### Comparison with Prompt-debias (Chuang et. al.)
> Thank you for recognizing a similar approach and the novelty of our work. We agree that while their goal in debiasing VLM embeddings is analogous to ours, our methodologies differ significantly.
>
> We include the experimental results of Prompt-debias in Table 1 of the rebuttal PDF file. It turns out that while Prompt-debias can mitigate bias in various downstream tasks, our method, **SFID, performs significantly better**. Moreover, in text-to-image generation, Prompt-debias works only with neutral prompts but deteriorates the bias in the case of mismatch rates for gender-prompts.
>
> The reason our method works better than Prompt-debias is due to the bias amplification in the decoder or image encoder. Prompt-debias's application is limited to the text embedding, which is the output of the text encoder. In contrast, SFID mitigates bias in either the encoder or decoder output, or both, providing a more comprehensive approach to debiasing.
>
> ### Visualization of embedding translation by SFID
> We include a visual aid for the embedding to indicate the bias-free representation achieved by SFID in **Figure 2 of our rebuttal PDF**. As SFID aims to obscure all the features in the important index by imputing ambiguous values, all the data points are gathered at a single point to mute the information related to the sensitive attributes, while maintaining the features in other indices as they are. It is important to note that although this translation is enforced to a single point, it still remains within the distribution of the original samples, as shown in Figure 3 of our main paper.
>
> ### Grammar and Structure
> Thank you for pointing out the organization of our paper. We will address the minor grammatical errors and ensure clarity throughout. In Section 3, it is true that it serves both as preliminaries and motivation section. Experimental setups are included in this section to provide readers with context regarding each downstream task and experimental setting, ensuring consistency. We will revise and reorganize our content to enhance readability.
>
> ### Limitation and Future work of SFID
> Thank you for pointing out the limitations section. We have identified potential limitations and areas for future exploration in **our global rebuttal.**

---

> > ### Comment · Reviewer_Wrj8 · 2024-08-12
> >
> > I thank the authors for their comprehensive rebuttal. I have raised my score accordingly.

---

### Official Review · Reviewer_umE5 · 2024-07-12

**Soundness:** 2
**Presentation:** 3
**Contribution:** 3
**Rating:** 6
**Confidence:** 3

**Summary:**

this paper introduces Selective Feature Imputation for Debiasing (SFID), which integrates feature pruning and low confidence imputation (LCI) to effectively reduce biases in VLMs.

**Strengths:**

1.The proposed method utilize feature selection techniques such as RandomForest to identify gender-specific (or race) biases within the frozen representation, and subsequently replace bias-causing features with bias-free representation.
2. SFID eliminates the need for costly retraining of pre-trained VLMs and it simply utilizes datasets with sensitive attributes in individual images or texts for debiasing.
3.The experimental results demonstrate the efficacy of the proposed method in mitigating bias across 4 downstream tasks

**Weaknesses:**

1.the method is simple, utilizing Random Forest, which is pretty well-known in the machine learning community. However, the novelty is a little limited.
2. how to define the \Delta DP for attributes with more than two levels? For instance, race has white, asian, black, etc.
3. what is the principle for choosing the number of important features? will it affect the performance?

**Questions:**

About the working mechanism of the proposed method, the authors may dig deep how the identified features is correlated to the social biases, maybe with visualization tools.

**Limitations:**

see weaknesses and questions

---

> ### Author Rebuttal · Authors · 2024-08-06
>
> ### Novelty of the proposed method
> We appreciate the reviewer's opinion regarding the simplicity and novelty of utilizing RandomForest in our framework.
>
>
> While the RandomForest algorithm itself is well-known and simple, the novelty of our work lies not in the use of RandomForest alone, but in the innovative way we integrate it within our Selective Feature Imputation for Debiasing (SFID) framework. The SFID methodology introduces several novelties:
> - **Selective Feature Imputation:**  Our approach combines feature pruning with low-confidence imputation (LCI) to replace bias-causing features with bias-free representations. This novel technique maintains the semantic integrity of the embeddings while effectively reducing bias.
> - **Efficiency and Cost-Effectiveness**: SFID eliminates the need for costly retraining of pre-trained VLMs and does not require paired text-image datasets, making it a cost-effective and efficient solution for debiasing.
> - **Versatility Across Modalities and Tasks**: Unlike existing methods that focus on specific modalities or tasks, SFID is designed to be seamlessly integrated into various components of VLMs, including both encoders and decoders, across a wide range of tasks such as zero-shot classification, text-to-image retrieval, image captioning, and text-to-image generation. **We emphasize that none of the existing work can debias such a seamless number of downstream tasks without further training or fine-tuning.**
> - **Empirical Validation and Effectiveness**: Our experimental results demonstrate the practical effectiveness of SFID in mitigating biases across various tasks without compromising performance. This is evidenced by significant improvements in fairness metrics across multiple benchmarks. The comparative analysis with other debiasing methods, such as DeAR, CLIP-clip, and Prompt-debias (newly added), highlights t**he superior performance of SFID**, further validating the novelty and utility of our approach.
>
> While RandomForest is indeed a well-known algorithm, the novelty of our work lies in the innovative application and integration of this algorithm within the SFID framework. This novel approach offers a significant advancement in the field of debiasing VLMs, as demonstrated by our empirical results and comparative analysis. Furthermore, we highlight the necessity for a unified debiasing framework, such as SFID, to address the challenges posed by the emergence of diverse VLMs applications.
>
> ### Multiple sensitive attributes in racial bias
> Thanks for pointing out the details how to define $\Delta DP$ in multiple attributes and how SFID mitigates such bias.
>
> SFID is free from the type of bias when the attribute labels are given such as gender and race in FairFace dataset, even in scenarios involving multiple attributes.
> We define $\Delta DP$ for multiple attributes by adopting the evaluation metric from [1] and [2], which measures the maximum discrepancy across the sensitive attributes for each class. This metric is defined as follows:
>
> \begin{align*}
> DP_c =  \max_{i \in S} \max_{j \in S \setminus \{i\}} \Bigg( \bigl| P(y=c \mid a=i) - P(y=c \mid a=j) \bigr| \Bigg)
> \end{align*}
> where $i, j \in S$, $c$ is a class, and $S$ denotes the set of multiple sensitive attributes. We consider the mean and maximum values of $DP_c$ to evaluate the bias across the classes. By applying this metric, we can effectively measure and demonstrate the reduction in bias achieved by SFID, despite the differences in sensitive attributes between the training and test sets.
>
> To support the adaptability of SFID in addressing different types of bias with multiple attributes, we report experimental results for racial bias in our **global rebuttal**. The table in the global rebuttal demonstrates that SFID can effectively mitigate racial bias even when dealing with multiple sensitive attributes. This showcases SFID's versatility and robustness in reducing bias across various scenarios.
>
> [1] Foulds, James R., et al. "An intersectional definition of fairness." 2020 IEEE 36th International Conference on Data Engineering (ICDE). IEEE, 2020.
> [2] Denis, Christophe, et al. "Fairness guarantee in multi-class classification." arXiv preprint arXiv:2109.13642 (2021).
>
> ### Impact of the number of important features
> Thank you for pointing out how to choose the number of important features. Indeed, the number of important features significantly affects performance, but we have an easy way to determine the optimal number, $k$, as shown in Appendix A.1 of the main paper.
>
> In SFID, selecting $k$ is crucial for performance. As shown in Figure 5 of Appendix A.1, we determine an appropriate value for $k$ by identifying the elbow point in the feature importance curve, obtained by sorting the indices by their importance. According to Figure 5, $k=50$ is a reasonable elbow point.
>
> We also analyze the impact of $k$ on performance. As expected, we achieve the best trade-off between utility and fairness when $k=50$, as shown in Figure 6 and Figure 7 in Appendix C of the main paper.
>
> Overall, we have a solid guideline for choosing the number of important features, $k$, and acknowledge its impact on performance. The observed trend aligns with our intuition.
>
> ### Visualization: Correlation of identified features to social biases
> Thank you for suggesting this valuable enhancement to our paper.
>
> We visualize how the important features are correlated to social biases by showing GradCAM visualizations, as presented in **Figure 1 of our rebuttal PDF file**. For example, the more important features highlight human faces, while the least important features are correlated with the image background.
>
> SFID imputes the representations at important indices with the values from low-confidence samples, making the face-related features ambiguous. Despite this imputation, the replaced values remain within the distribution of the original samples, as shown in Figure 3 of our main paper.

---

> > ### Comment · Reviewer_umE5 · 2024-08-13
> >
> > I thank the authors for their comprehensive rebuttal, which has addressed my concerns. So I have raised my score accordingly. Please include the important analysis and definition in the main content.

---

### Official Review · Reviewer_ystz · 2024-07-17

**Soundness:** 3
**Presentation:** 3
**Contribution:** 3
**Rating:** 7
**Confidence:** 4

**Summary:**

The paper introduces a new method to reduce biases in VLMs, which works by using a random forest to identify the bias-related features in model representations and then imputes the values for those features with values from the low-confidence samples. The authors test it on tasks like zero-shot classification, image captioning, and text-to-image generation.

**Strengths:**

- The method works across different tasks (classification, captioning, generation) and model components (encoders, decoders).
- SFID doesn't require costly model retraining or extensive hyperparameter tuning.
- The authors test SFID on multiple state-of-the-art VLMs (CLIP, XVLM) and compare it against existing debiasing methods.

**Weaknesses:**

- The paper focuses primarily on gender bias without exploring how the method performs on more complex biases, or even biases on other attributes -- e.g. race or age attributes are provided in the FairFace dataset, it would be good to show how SFID can mitigate biases w.r.t these attributes.
- The paper doesn't address potential limitations of using Random Forest for feature importance. There could be cases where complex, non-linear relationships between features and bias are missed by this approach.
- Insufficient comparison with a wider range of existing debiasing methods. The paper only compares SFID with DeAR and CLIP-clip, and is potentially missing out on how it compares to other methods e.g. https://arxiv.org/pdf/2302.00070.
- Notation in Section 3 could be simplified (especially 3.3)

**Questions:**

- In line 188 the authors mention that “For a frozen component in VLMs g, whether it is an encoder or decoder, or processes image or text, we obtain the frozen representations”. Can the authors provide a breakdown of the computation cost when the encoder or the decoder is a language model that is large?
- Would it be possible to generate embedding translation (figure 4) for SFID as well?

**Limitations:**

The authors have provided some limitations in section 5.2. The paper could benefit from an extended limitations section.

---

> ### Author Rebuttal · Authors · 2024-08-06
>
> ### How SFID Mitigates Bias in Various Attributes, Including Racial Bias
> Thank you for your detailed observations on how SFID mitigates different types of bias, which is even more complex.
>
> SFID effectively addresses biases when attribute labels are provided such as gender and race in the FairFace dataset. The RandomForest attribute classifier is trained to identify important features within the frozen image embedding. For instance, as illustrated in **Figure 1 of our rebuttal PDF**, GradCAM highlights that the 'important features' pertain to human faces, allowing for the recognition of identity in both gender and race attribute cases. SFID aims to impute the values in these important features with those of low-confidence samples, rendering the final embedding ambiguous in terms of the targeted attribute.
>
> To demonstrate the adaptability of SFID to different types of attributes, we have included experimental results addressing racial bias in our **global rebuttal**. Both **Figure 1 in the PDF** and the **Table in the global rebuttal** illustrate SFID's capability to mitigate more complex biases, showcasing its overall effectiveness.
> ### Effectiveness in using RandomForeset
> We appreciate the reviewer's insightful feedback regarding the potential limitations of using RandomForest for feature importance. However, we argue that using RandomForest for feature importance is both theoretically and empirically sufficient and effective.
> - Contrary to some concerns, RandomForest is capable of capturing complex, non-linear relationships due to its ensemble nature. By combining multiple decision trees, each considering different feature subsets and data splits, RandomForest can effectively model complex interactions and dependencies among features.
> - Specifically, given the ample representation of images or text extracted by pre-trained neural networks, RandomForest's role is to identify important features while considering the dependencies between them.
> - In short, RandomForest is an interpretable method for determining feature importance, robust against overfitting, and efficient in handling large datasets.
> - Moreover, the empirical results in the paper demonstrate the effectiveness and versatility of using RandomForest within our framework.
>
> While we have demonstrated the effectiveness of RandomForest in our current work, we acknowledge the importance of exploring additional techniques that might further enhance the modeling of complex relationships. Future research could incorporate advanced methods for identifying important features while improving the quality of representation.
> ### Comparison beyond DeAR and CLIP-clip
> Thank you for your feedback. We acknowledge that many debiasing methods have been proposed recently, but most focus on specific downstream tasks, with only a few demonstrating versatility across various tasks, such as DeAR and CLIP-clip. The paper suggested by the reviewer, Prompt-debias [1], is limited to scenarios where the VLM input is text, making it inapplicable to tasks such as image captioning.
>
> For the other three tasks, we have reported the results in **our rebuttal PDF file**. The results in Table 1 show that while Prompt-debias mitigates bias in zero-shot classification and text-to-image retrieval, ***our method, SFID, is superior in mitigating bias in these tasks compared to Prompt-debias***. In the case of text-to-image generation, Prompt-debias shows slightly better performance with neutral prompts with CoDi, but significantly deteriorates the bias with gender prompts.
>
> In conclusion, few methods are versatile across various downstream tasks. SFID demonstrates superior performance in mitigating various types of bias across all the downstream tasks we evaluated.
>
> [1] Chuang, Ching-Yao, et al. "Debiasing vision-language models via biased prompts." arXiv preprint arXiv:2302.00070 (2023).
> ### Notation in Section 3
> Thank you for pointing out the readability issues. We will revise the notations to be simpler and will add an appendix to provide more detailed explanations and visualizations for each evaluation metric to increase readability.
> ### Computational Cost
> We are happy to address the computational cost for each component and data type. **The overall computational cost is significantly low** since SFID only utilizes frozen embeddings from a pre-trained network, without involving any training or fine-tuning of the neural network. Regardless of the network size, we break down the computational cost into two parts: a) identifying feature importance and b) inference. Both tables demonstrate that applying SFID does not increase the computational cost in practice.
> #### Identifying Feature Importance (One-time for each component)
> |Component|Details|
> |-|-|
> |CLIP Encoders|54.60s to 60.75s|
> |CoDi Text Encoder|65.90s|
> |CoDi Image Decoder|104.14s|
> #### Inference
> |Component|Details|
> |-|-|
> |Zero-shot classification|0.196ms / sample|
> |Zero-shot classification + SFID| 0.204ms / sample|
> |T2I retrieval | 0.146s / sample|
> |T2I retrieval + SFID| 0.152s / sample|
> |T2I generation (CoDi)| 11.80s / 25 prompts|
> |T2I generation (CoDi) + SFID | 12.05s / 25 prompts|
> ### Visualization of embedding translation by SFID
> We include a visual aid for the embedding to indicate the bias-free representation achieved by SFID in **Figure 2 of our rebuttal PDF**. As SFID aims to obscure all the features in the important index by imputing ambiguous values, all the data points are gathered at a single point to mute the information related to the sensitive attributes, while maintaining the features in other indices as they are. It is important to note that although this translation is enforced to a single point, it still remains within the distribution of the original samples, as shown in Figure 3 of our main paper.
> ### Limitation and Future work of SFID
> Thank you for pointing out the limitations section. We have identified potential limitations and areas for future exploration in **our global rebuttal.**

---

> > ### Comment · Reviewer_ystz · 2024-08-13
> >
> > Thanks for the comprehensive response. I have raised my score accordingly.

---

### Author Rebuttal · Authors · 2024-08-06

Thank you to the reviewers for the valuable feedback! Here is our global rebuttal, with a PDF file attached for figures and tables. Please refer to the individual rebuttals for specific details and additional information.

### SFID mitigates various types of bias, even in multi-attribute case
To address the reviewers' concerns regarding various types of bias in Vision-Language Models (VLM), especially in more complex cases involving multiple sensitive attributes, we conducted additional experiments focusing on racial bias with more than two sensitive attributes.

Firstly, we adopted the FairFace dataset for training the attribute classifier, as it contains seven racial attributes: East Asian, Indian, Black, White, Middle Eastern, Latino Hispanic, and Southeast Asian. Given that RandomForest can handle multiple classes, SFID is also applicable in this context. During the evaluation stage for zero-shot classification, we used the FACET dataset, which contains 'skin tone' labels instead of race. We categorized the skin tone attributes into three categories: 'lighter,' 'middle,' and 'darker.' In this setting, the biased zero-shot classifier tends to produce higher accuracy by associating certain skin tones with specific professions (e.g., bartender with lighter skin, trumpeter with darker skin).

To mitigate this bias, SFID demonstrates its effectiveness even though the attributes in the training set and test set do not exactly match, i.e. race in FairFace and skin tone in FACET. We adopted an evaluation metric inspired by [1] and [2], which measures the maximum discrepancy across the sensitive attributes for each class. This metric is defined as follows:

\begin{align*}
DP_c =  \max_{i \in S} \max_{j \in S \setminus \{i\}} \Bigg( \bigl| P(y=c \mid a=i) - P(y=c \mid a=j) \bigr| \Bigg)
\end{align*}
where $i, j \in S$, $c$ is a class, and $S$ denotes the set of multiple sensitive attributes. We consider the mean and maximum values of $DP_c$ to evaluate the bias across the classes. By applying this metric, we can effectively measure and demonstrate the reduction in bias achieved by SFID, despite the differences in sensitive attributes between the training (debiasing) and test sets.

| Method | Mean Acc. | Mean DP | Max DP |
| -------- | -------- | -------- | -------- |
| CLIP-RN50 (Baseline)     | 51.92     | 13.94     | 33.89     |
| CLIP-RN50 (SFID)     | 51.35     | **13.27**     | **32.56**     |
| CLIP-ViT-B/32 (Baseline)     | 52.48     | 13.54     | 44.62     |
| CLIP-ViT-B/32 (SFID)    | 51.97     | **13.31**     | **32.71**     |
| XVLM (Baseline)     | 56.61     | 14.85     | 45.30     |
| XVLM (SFID)    | 56.51     | **14.59**     | **43.00**     |

The results show that even with limited data and racial bias present only in the image dataset, SFID can effectively mitigate the racial bias. We expect even more advanced results if we have access to race-profession related text datasets as well.

[1] Foulds, James R., et al. "An intersectional definition of fairness." 2020 IEEE 36th International Conference on Data Engineering (ICDE). IEEE, 2020.
[2] Denis, Christophe, et al. "Fairness guarantee in multi-class classification." arXiv preprint arXiv:2109.13642 (2021).



### Limitation and future work of SFID
In response to the reviewer's concerns, we have identified some limitations of SFID and potential areas for future exploration.
##### Dependence on validation set
SFID assumes that labels in the debiasing dataset (training and validation set) are available. If the validation set is unavailable, a subset of the training set could be used as a validation set. A potential limitation is that the quality of low-confidence samples depends on the representativeness of the validation set. For example, if the samples in the validation set are out-of-distribution, the imputed values will follow this distribution, potentially worsening performance.
##### Compound bias
SFID can be applied to various types of bias, as shown in our response above. However, there is room for improvement in mitigating more than one type of bias simultaneously, such as gender and race together. As future work, we plan to extend our framework to address compound bias.

##### Diversity in Low Confidence Imputation
In the Low Confidence Imputation (LCI) process, we take the average of all low-confidence samples for each feature index. Although the imputed value remains in-distribution, the diversity of the imputed value could be limited as they are projected into a single imputation value. We will explore how the distribution of the imputation value affects diversity and how to properly distribute the imputation values to ensure diversity in the generation task while maintaining debiased results.

We appreciate the reviewer's insights and will address these limitations in our future work.

---

### Comment · Area_Chair_BUfr · 2024-08-12
**Please read the author rebuttal, other reviews and respond to the authors NOW!**

Dear Reviewers,

Thanks to those of you who already responded to the authors acknowledging the rebuttal and asking follow-up questions if any.

Those who have not responded yet, please do the following ASAP: thoroughly read the rebuttal, the other reviews and respond to the authors about whether all your questions / concerns have been addressed or not. If not, please elaborate on which questions / concerns are still not addressed so that the authors have fair chance of addressing them before the author-reviewer discussion period ends in ~41 hours from now (August 13th, 11:59pm AoE).

Your AC

---

### Decision · Program_Chairs · 2024-09-25

**Decision:**

Accept (spotlight)

**Comment:**

The reviewers find the problem of debiasing of VLMs being studied in this paper to be timely and important, the proposed method to be effective, lightweight and easy to apply, and the experimental results to be convincing. The reviewers appreciate the fine-tuning free nature of the proposed method and the fact that it works across multiple tasks and model components.

The reviewers had raised some concerns, but the rebuttal successfully addressed most of them and all reviewers recommend acceptance. The authors are recommended to improve the final paper version by following the reviewer recommendations.